# Exposure of Rats to Multi-Walled Carbon Nanotubes: Correlation of Inhalation Exposure to Lung Burden, Bronchoalveolar Lavage Fluid Findings, and Lung Morphology

**DOI:** 10.3390/nano13182598

**Published:** 2023-09-20

**Authors:** Tatsuya Kasai, Shoji Fukushima

**Affiliations:** Japan Bioassay Research Center (JBRC), Japan Organization of Occupational Health and Safety, Hadano 257-0015, Japan

**Keywords:** carbon nanotubes, MWCNT, MWNT-7, lung burden, carcinogenicity

## Abstract

To evaluate lung toxicity due to inhalation of multi-walled carbon nanotubes (MWCNTs) in rats, we developed a unique MWCNT aerosol generator based on dry aerosolization using the aerodynamic cyclone principle. Rats were exposed to MWNT-7 (also known as Mutsui-7 and MWCNT-7) aerosolized using this device. We report here an analysis of previously published data and additional unpublished data obtained in 1-day, 2-week, 13-week, and 2-year inhalation exposure studies. In one-day studies, it was found that approximately 50% of the deposited MWNT-7 fibers were cleared the day after the end of exposure, but that clearance of the remaining fibers was markedly reduced. This is in agreement with the premise that the rapidly cleared fibers were deposited in the ciliated airways while the slowly cleared fibers were deposited beyond the ciliated airways in the respiratory zone. Macrophage clearance of MWNT-7 fibers from the alveoli was limited. Instead of macrophage clearance from the alveoli, containment of MWNT-7 fibers within induced granulomatous lesions was observed. The earliest changes indicative of pulmonary toxicity were seen in the bronchoalveolar lavage fluid. Macrophage-associated inflammation persisted from the one-day exposure to MWNT-7 to the end of the two-year exposure period. Correlation of lung tumor development with MWNT-7 lung burden required incorporating the concept of area under the curve for the duration of the study; the development of lung tumors induced by MWNT-7 correlated with lung burden and the duration of MWNT-7 residence in the lung.

## 1. Introduction

Carbon nanotubes (CNTs) are a new material with a cylindrical honeycomb structure (graphene) in which carbon atoms are hexagonally bonded. CNTs are broadly divided into single graphene cylinders (SWCNTs) with diameters of 0.7–3 nm and multi-walled CNTs (MWCNTs) composed of two or more concentric cylinders with diameters of 10–200 nm [1]. SWCNTs have a very small diameter and large surface area, which increases the van der Waals forces, and they tend to cluster together in rope-like bundles. Depending on the number of layered graphene cylinders, MWCNTs are in aggregated, curly, or acicular structures with varying lengths.

Since many MWCNTs are light weight, show an asbestos-like needle structure, and can be easily airborne as aerosols, the health effects due to inhalation for workers who handle MWCNTs has been a concern since before commercial production began. The National Institute of Health Sciences, Tokyo, Japan, investigated the carcinogenic hazard of the MWCNT known as MWNT-7 (also known as Mitsui-7 and MWCNT-7). Their findings, published in 2008, reported that mesothelioma was induced by intraperitoneal injection of MWNT-7 in p*53* heterozygous mice [2]. This initial study was subject to criticism of the preparation of the particles injected, the dose administered, and the mouse model used for the study [3,4]. However, another study, conducted by the Tokyo Metropolitan Institute of Public Health, subsequently reported mesothelioma development was caused by intrascrotal injection of MWNT-7 in Fischer 344 rats [5].

In consideration of the fact that cancer is one of the most devastating occupational health problems, the Ministry of Health, Labour and Welfare of Japan contracted the Japan Bioassay Research Center to conduct studies on whether inhalation exposure to MWCNTs can lead to the development of lung cancer and mesothelioma. Given the results of the studies by Takagi et al. [2] and Sakamoto et al. [5] and the fact that the fiber shape of MWNT-7 is similar to that of asbestos fibers, MWNT-7 was selected as the test material. In order to conduct a long-term inhalation exposure carcinogenicity study, an inhalation exposure system able to expose test animals to MWNT-7 for 6 h a day for 2 years needed to be developed. We developed a unique MWNT-7 aerosol generator based on dry aerosolization using the aerodynamic cyclone principle. This apparatus separates MWNT-7 aggregates and fibers by centrifugal force and gravitational sedimentation, allowing only MWNT-7 fibers able to pass through a partition sieve to be transported into the inhalation chamber. Size-limited MWNT-7 aerosols were generated for each target concentration. Our initial report on the development of this system and the results of our first study, a 6 h exposure of male rats to aerosolized MWNT-7, was submitted for publication in November 2012, although publication was delayed until 2014 [6].

The results of our 1-day, 2-week, and 13-week studies demonstrated that inhalation of MWNT-7 resulted in acute inflammation, granulomatous change, and focal fibrosis in the lung [6,7,8]. The 13-week study also indicated that prolonged inhalation of MWNT-7 would very likely result in induction of lung tumors and possibly pleural mesothelioma. However, these short-term toxicity studies did not directly demonstrate that inhalation of MWNT-7 represented a carcinogenicity hazard. However, another study by Sargent et al. demonstrated a possible hazard [9]. This study used an initiation–promotion protocol in which mice were first treated with 3-methylcholanthrene to initiate neoplastic transformation of lung cells and then exposed to MWNT-7 by whole-body inhalation for 15 days. Sargent et al. reported that inhalation exposure of mice to MWNT-7 promoted the neoplastic progression of 3-methylcholanthrene initiated lung cells. Based in large part on these experimental data and the intrascrotal injection study by Sakamoto et al., and on the intraperitoneal studies by Nagai et al. and Takagi et al. [5,9,10,11], IARC determined that there was sufficient evidence in experimental animals for the carcinogenicity of MWNT-7 (also known as MWCNT-7), and classified MWNT-7 as possibly carcinogenic to humans (Group 2B) in 2014 [12]; the final report was published in 2017 [13] (see p. 192).

Although the IARC classification formalized the identification of the carcinogenicity hazard of MWNT-7, there was still no direct evidence that administration of MWNT-7 by a physiologically relevant route would induce cancer in experimental animals. Our 2-year whole-body inhalation study showed clear evidence that inhalation of MWNT-7 was carcinogenic to the lungs of male and female F344 rats [14]. Demonstration of the carcinogenicity of inhaled MWNT-7 in rats validated the classification of MWNT-7 as a Group 2B carcinogen. Importantly, this study used a pathway of exposure to MWNT-7 that can be extrapolated to humans, inhalation.

This paper comprehensively reviews our previously reported 1-day, 2-week, 13-week, and 2-year inhalation exposure studies that were conducted using this device [6,7,8,14,15], along with additional unpublished data. Our aim is to provide new insights into the toxicity, especially the carcinogenicity, of MWNT-7.

## 2. Materials and Methods

All of our studies were conducted in accordance with the Organization for Economic Cooperation and Development (OECD) inhalation toxicity testing guidance document TG-39 and in compliance with the OECD guidelines for each time period of exposure. In particular, the carcinogenicity study was conducted in accordance with the “Standards to be observed by testing laboratories” of Notification No. 76 of the Ministry of Labor, 1 September 1988 (revised, Notification No. 13 of the Ministry of Labor, 29 March 2000), and was conducted in accordance with OECD Good Laboratory Practice (OECD, 1997), and with reference to OECD Test Guideline for Testing of Chemicals 451 “Carcinogenicity Studies” (OECD, 2009). All animal experiments were also conducted in accordance with the “Guidelines for the Proper Conduct of Animal Experiments” (Science Council of Japan, 2006), and were approved by the ethics committee of the Japan Bioassay Research Center (JBRC).

### 2.1. Test Materials

The MWCNTs were purchased from Hodogaya Chemical Co., Ltd. (MWNT-7, Lot No. 071223 and 080126, Tokyo, Japan), and used without further purification, sieving, or sonication. According to the information provided by the manufacturer, the MWNT-7 was generated with a floating chemical vapor deposition (CVD) process, with a carbon purity > 99.6% (Lot No. 071223) and >99.8% (Lot No. 080126), a nominal mean diameter of 40–90 nm, an aspect ratio of >100, and a surface area of 24–28 m^2^/g.

Lot No. 080126 MWNT-7 was used for two single (one day) exposure studies and a two-week exposure study. One of the one-day exposure studies was for aerosol generator development and functional verification, and the other was for analytical method development and analytical verification. Both lot No. 071223 and 080126 were used for the 13-week study and for the 2-year study (carcinogenicity study).

We measured the actual diameter and length of the bulk MWNT-7: Lot No. 080126 fibers had an average width of 90.7 nm and length of 5.7 μm, with 48.7% of the tubules being longer than 5 μm [6]. Lot No. 071223 fibers had an average width of 83.8 nm and length of 5.2 μm, with 45.1% of the tubules being longer than 5 μm [14].

### 2.2. Animals and Breeding Environment

F344/DuCrlCrlj (SPF) rats were used in all studies. The rats were obtained at 4 weeks of age from Charles River, Japan (Kanagawa, Japan) and exposed to MWNT-7 after 2 weeks of quarantine and acclimation. Only male rats were used in the 1-day study, while male and female rats were used in the other studies. After quarantine and acclimation, the 37 male rats used for the 1-day exposure study [6], the 40 rats of each sex used for the 2-week study [7], the 40 rats of each sex used for the 13-week study [8], and the 200 rats of each sex for the 2-year carcinogenicity study [14] were assigned by stratified randomization to weight-matched MWNT-7-exposure and control clean-air-exposure groups.

Using dedicated whole-body exposure chambers, rats were exposed to MWNT-7 for 6 h per day for all exposure durations (1 day, 2 weeks, 13 weeks, and 2 years) according to the guidelines stated above. The animals were housed individually in stainless-steel wire hanging cages in stainless-steel inhalation exposure chambers. The exposure chambers (one for each MWNT-7 exposure level and one for the clean-air control) were in barrier-maintained animal rooms. The chamber environment was maintained at a temperature of 23 ± 2 °C and relative humidity 55 ± 10% in all studies, with 10 or 12 air changes per hour during the exposure periods. Fluorescent lighting in the animal room was automatically controlled to give a 12 h light/dark cycle. All rats in the 1-day, 2-week, and 13-week studies were fed ad libitum on commercial pellet feed (CRF-1, Oriental Yeast Co., Ltd., Tokyo, Japan) and sterile tap water except during the time of exposure. In the 2-year study, throughout the experimental period, all animals had free access to sterilized water and pellet diet (CRF-1).

### 2.3. Development of the Generator Used for Inhalation Exposure to MWNT-7 Aerosols

We developed an inhalation exposure system (cyclone sieve method) that keeps the MWNT-7 inhalation exposure concentration uniform throughout the exposure period and keeps the particle size the same for each target concentration [6].

The layout of the aerosol generation and inhalation exposure system for MWNT-7 is shown in Figure 1 (adapted from Figure 1 Kasai et al. [6]). Our system (cyclone sieve method) is a dry aerosolization method based on the aerodynamic cyclone principle. Bulk MWNT-7 is introduced into the middle portion of the generator by the dust feeder. Clean filtered air is introduced into the cyclone vessel below the dust feeder and a high-speed upward spiraling airstream is continuously generated. Bulk MWNT-7 is dispersed and aerosolized by collision with the airstream. Light MWNT-7 particles are carried upward to the partition sieve (diameter 21 cm, mesh size 53 μm) that is installed at the top of the cyclone vessel. Only MWNT-7 fibers that pass through the sieve will reach the inhalation chambers. MWNT-7 fibers that are not dispersed and aerosolized remain in the middle position of the cyclone vessel and are removed from the cyclone vessel by pulling them in the downstream direction with a constant volume of air, the exhaust. Since MWNT-7 is easily agglomerated by static electricity, multiple static electricity eliminators (ionizers) are installed to prevent fiber agglomeration. This system generates single fibers of MWNT-7 of relatively uniform size. 

The fibers’ mass-per-fiber number in the air samples from the inhalation chambers was determined periodically throughout each study; the number of times the air was sampled is specified in the Methods section of each study. The fiber number concentration in the inhalation chambers was continuously monitored using an optical particle controller (OPC). Using the fiber number-to-mass measurement and the concentration of fibers in the chamber, the fiber mass concentration (mg/m^3^) in the chamber was calculated. The aerosol concentration in the chamber was controlled by feedback control to the dust feeder using the OPC upper and lower mass concentration limit signals, i.e., when the aerosol concentration in the chamber rose above the upper limit of the designated concentration range, the OPC signaled the dust feeder to stop supplying MWNT-7 to the aerosol generator/sieving unit, and when the aerosol concentration in the chamber fell below the lower designated concentration range, the OPC signaled the dust feeder to resume supplying MWNT-7 to the aerosol generator/sieving unit. 

### 2.4. Lung Burden Analysis

Lung deposition was analyzed using a new method using hybrid markers developed by Ohnishi et al. [15]. The principle of this analytical method is based on a correlation between MWNT-7s and benzo[*ghi*]perylene (B(ghi)P) adsorbed onto the MWNT-7 fibers. Although MWNT-7s cannot be measured directly with this method, quantitative evaluation of MWNT-7s is possible by adsorbing B(ghi)P onto known amounts of MWNT-7 fibers and measuring the amount of marker after desorption by HPLC using fluorescence spectroscopy, thereby generating a standard curve. The ability to measure MWNT-7s with extremely high sensitivity and the ability to perform repeatable analyses was validated, enabling quantitative evaluation of MWNT-7s with high accuracy.

The actual procedure is briefly described here. Lung tissue (0.04–0.12 g) is dissected from the right or left lung, and the tissue is completely digested according to the method of Kohyama et al. [16]. The supernatant (12,000 rpm, 10 min) is filtered and dried. A known amount of B(ghi)P is added to the fibers and adsorbed onto the dispersed MWNT-7s. Excess B(ghi)P is removed, and the adsorbed B(ghi)P is desorbed with an organic solvent and quantified by HPLC. The amount of MWNT-7s is calculated using a standard curve, as mentioned above. 

### 2.5. Bronchoalveolar Lavage Fluid (BALF) and Pathological Examinations

BALFs were collected under anesthesia at the time of terminal necropsy after overnight fasting. In the 1-day, 2-week, and 13-week exposure studies, the BALF was collected from the right lung by lavage with physiological saline solution. For the 2-year study, the left lung was lavaged with Eagle’s Minimum Essential Medium (MEM, adjusted to pH 7.2 with 1 N NaOH and 15 mM HEPES without sodium hydrogen carbonate). 

Biochemical examination of the BALF measured albumin (1-day, 2-week, and 13-week studies) and total protein (TP) (2-week, 13-week, and 2-year studies) as an indicator of alveolar–capillary permeability. Lactate dehydrogenase (LDH) was measured in the 1-day, 13-week, and 2-year studies, and alkaline phosphatase (ALP) was measured in the 2-week, 13-week, and 2-year studies as indicators of alveolar cytotoxicity.

In the cytological examination of the BALF, total cells were counted using automatic cell analyzers. In addition, neutrophils, lymphocytes, and alveolar macrophages were counted under a light microscope. At least 500 cells were counted, and the number of cells/unit volume was determined. Macrophages were also morphologically characterized and classified. 

All organs and tissues were examined histopathologically. The organs and tissues were fixed in 10% neutral buffered formalin. Right or left lungs that were not used for BALF collection were directly fixed by immersion in 10% neutral buffered formalin. Fixed tissues were embedded in paraffin and 3–5 μm thick sections were prepared and stained with hematoxylin and eosin (H&E). Lung sections were also stained with Kernechtrot stain (1-day, 2-week, and 2-year studies) or with Masson’s trichome stain and carefully examined. Histopathological diagnosis was performed by pathologists certified by the Japanese Society of Toxicologic Pathology, and in the 2-year carcinogenicity study histopathological diagnosis was peer reviewed by outside pathologists. 

### 2.6. Summary of Studies Used to Complete the 2-Year Carcinogenicity Study

(A) The initial study undertaken was verification of aerosol generation and confirmation of exposure of rats to MWNT-7 [6]: 

(A-1) Performance test of generator for 6 h at 0.2, 1, and 5 mg/m^3^ without animals.

(A-2) Verification using rats (final test): 0 and 5 mg/m^3^, 6 h (one day) exposure. 

BALF and pathological examination was conducted on day 1, day 7, and day 28 after exposure.

This study confirmed the capability of the whole-body inhalation exposure system to generate MWNT-7 aerosols at specified concentrations. 

(B) A method to measure the amount of fibers present in lung tissue (lung burden) was developed [15]. This study used rats intratracheally administered 2 μg MWNT-7 per rat and rats exposed by inhalation to 5 mg/m^3^ MWNT-7 for 6 h (one-day exposure): 

(B-1) Measurement of MWNT-7 in the lung tissue immediately after intratracheal instillation and immediately after completion of the 6 h inhalation exposure.

(B-2) Measurement of clearance on day 1, 7, 28, and 56 after exposure.

This study verified the capability of this method to accurately measure MWNT-7 lung burden.

(C) A two-week toxicity study was undertaken using inhalation exposure to 0, 0.2, 1, and 5 mg/m^3^ MWNT-7 [7]. There was a total of 10 exposures: 6 h/day, 5 day/week, and 2 weeks (excluding Saturdays and Sundays). Lung burden analysis, examination of the BALF, and pathological examination were performed.

This study confirmed the capability of the whole-body inhalation exposure system to generate MWNT-7 aerosols at the same specified concentrations for 10 exposures over the course of two weeks.

(D) A thirteen-week toxicity study was undertaken using inhalation exposure of 0, 0.2, 1, and 5 mg/m^3^ MWNT-7 [8]. There was a total of 62 exposures: 6 h/day, 5 day/week, and 13 weeks (excluding Saturdays, Sundays, and national holidays). Lung burden analysis, examination of the BALF, and pathological examinations were performed.

This study confirmed the suitability of the whole-body inhalation exposure system for sub-chronic toxicity studies.

(E) A two-year carcinogenicity study was undertaken using inhalation exposure to 0, 0.02, 0.2 and 2 mg/m^3^ MWNT-7, based on the results of 13-week toxicity study [14]. There was a total of 492 exposures: 6 h/day, 5 day/week, and 2 years (excluding Saturdays, Sundays, and national holidays). Lung burden analysis, examination of the BALF, and pathological examinations were performed.

This study confirmed that inhalation exposure to MWNT-7 was carcinogenic in rats.

### 2.7. Statistics

The correlation graphs shown in Figures 3–5 were created using Microsoft Excel version 365, and all correlation coefficients (Table 2) were generated using Microsoft Excel version 365.

The incidences of lung carcinomas shown in Figure 10 were analyzed for significant difference from the clean-air-exposed group using Fisher’s exact test.

## 3. Results

### 3.1. MWNT-7 Fibers in the Lung Tissue

During the exposure periods described in Section 2.6, from 1 day to 104 weeks, the variations in mass concentration (CV; coefficient of variation) at each target concentration were all found to be within 12% of the target concentrations. In addition, the micro-orifice uniform deposition cascade impactor confirmed that the size distribution of MWNT-7 fibers had a mass median aerodynamic diameter (MMAD) of approximately 1.3 μm (maximum 1.6 μm) for all exposure periods, with a geometric standard deviation of approximately 3.0 for all exposure periods. 

In the 1-day exposure study [6], bulk MWNT-7 fibers were examined using a scanning electron microscope (SEM). The widths of 500 MWNT-7 fibers and the lengths of 1200 fibers were measured. The bulk MWNT-7 fibers had a mean width of 90.7 nm (median 88.8 nm) and a mean length of 5.7 μm (median 4.8 μm). The MWNT-7s collected from the 5 mg/m^3^ MWNT-7 chamber (see Section 2.6, A-1) had a mean width of 130.5 nm (median 91.8 nm) and a mean length of 5.7 μm (median 4.8 μm), similar to the width and length of the bulk MWNT-7. The MWNT-7s collected from the rat lung tissue immediately after the 6 h exposure had a mean width of 100.3 nm (median 88.1 nm) and a mean length of 8.1 μm (median 6.8 μm). Notably, the lung tissue which was used to obtain the MWNT-7 fibers included ciliated airways as well as the alveolar region. Consequently, the longer length of the fibers collected from the lung tissue compared to the lengths of the aerosolized MWNT-7 fibers in the inhalation chamber is likely due to higher ciliary transport out of the lung of shorter fibers compared to longer fibers. This suggests that the retention of inhaled MWNT-7s in the lung and their lung toxicity is influenced by their size. 

The 13-week study [8] used MWNT-7 lot No. 071223 and 080126. The mean width and length of the MWNT-7 fibers collected from the inhalation chambers at the three exposure concentrations were 94.1–98.0 nm and 5.5–6.2 μm. The 2-year carcinogenicity study [14] used the same two lots as the 13-week study. The mean width and length of the MWNT-7 fibers collected from the inhalation chambers ranged from 92.9 to 98.2 nm and 5.4 to 5.9 μm. Thus, there were no differences in the widths or lengths of MWNT-7s collected from the inhalation chambers at the mass concentrations used throughout the study periods. SEM and pathological examination confirmed the presence of MWNT-7 fibers in the lungs, the pleural and abdominal lavage fluids, and the mediastinal lymph nodes [8,14]. 

In the 2-year study, the MWNT-7 fibers found in the lung had an average width of 95.5–109.6 nm and length of 5.8–5.9 μm. In addition to numerous individual fibers, cocoon-like masses with a diameter of approximately 10–20 µm formed by MWNT-7 fibers were observed. The fibers wound around each other to form dense aggregates. Lysis of macrophages demonstrated that these MWNT-7 fiber aggregates were the result of phagocytosis of numerous fibers by a single alveolar macrophage. Notably, these cocoon-like masses were observed in the lungs of rats exposed to the lowest MWNT-7 concentration of 0.02 mg/m^3^, and they were observed in the lungs in an MWNT-7 exposure concentration-dependent manner.

Re-examination of the lung samples from the carcinogenicity study revealed masses of MWNT-7s that were 25–50 μm in diameter, larger than previously reported in the carcinogenicity paper [14]. Close examination of the tissue of the undigested samples revealed what appeared to be a collection of many small masses (Figure 2A). When such samples were dissolved a number of smaller cocoon-like masses of fibers were observed (Figure 2B). This indicates that the small masses shown in Figure 2B were phagocytosed MWNT-7s and that the collection of masses shown in Figure 2A is an aggregation of several macrophages.

Since the concentrations of MWNT-7 that the rats were exposed to in the inhalation studies were strictly controlled, from the one-day exposure study to the two-year exposure study, we hypothesized that the lung uptake of MWNT-7 in the absence of respiratory abnormalities would be related to exposure concentration and the duration of exposure. If so, it would be possible to quantitatively compare the lung burden after different exposure durations. In addition, in the lungs, macrophages would phagocytose the fibers, and this would be a factor in the effect that the fibers had on the lungs. Lung burden, BALF analysis, and pathological examinations are described below.

### 3.2. Lung Burden

Ohnishi et al. measured the MWNT-7 deposited in the left lung in five male rats immediately after a 6 h (1 day) exposure to 5 mg/m^3^ MWNT-7 and at 1 day, 7 days, 28 days, and 56 days after exposure [15]: 2.91 ± 0.23 μg immediately after exposure, 1.53 ± 0.27 μg the next day, 1.35 ± 0.34 μg after 7 days, 1.24 ± 0.15 μg after 28 days, and 1.20 ± 0.21 μg after 56 days (see Figure 5, Ohnishi et al. [15]). These values can be converted to whole-lung deposition using the weights of the left lung and the weights of the whole lung. MWNT-7 deposition per whole lung was 8.7 ± 0.6 µg immediately after exposure, 4.6 ± 0.8 μg the next day, 4.1 ± 1.0 µg after 7 days, 3.7 ± 0.5 µg after 28 days, and 3.6 ± 0.6 μg after 56 days. Compared to the lung burden immediately after exposure, approximately 47% of the MWNT-7s were cleared from the lungs by the next day. Alveolar clearance was modest, with approximately 88% after 7 days, 81% after 28 days, and 78% after 56 days of the MWNT-7 remaining in the lung based on next-day values. 

In the 2-week study [7], the mean lung burden one day after the end of the exposure period in males and females was 2.3 ± 0.4 and 2.2 ± 0.4 µg for the 0.2 mg/m^3^ group, 9.9± 1.2 and 8.7± 2.3 µg for the 1 mg/m^3^ group, and 43.4 ± 9.3 and 32.5 ± 9.7 µg for the 5 mg/m^3^ (Table 1). In agreement with the retention of fibers in the lung after a single 6 h exposure to MWNT-7, the lung burden in males exposed to 5 mg/m^3^ was 41.2 µg/lung at the end of the 4-week post-exposure period (see Discussion, Umeda et al. [7]).

In the 13-week exposure study, the mean fiber burden in the left lung one day after the end of the exposure period in males and females exposed to 0.2, 1, and 5 mg/m^3^ was 3.23, 21.2, and 120.3 mg/left lung in males and 2.30, 13.7, and 80.3 mg/left lung in females [8]. The weight ratio of the left lung to the right lung, 1:2, can be used to convert the lung burden of the left lung to the total lung burden: total lung burdens are shown in Table 1.

The total lung burden one day after the end of the exposure period in male and female rats exposed to 0.02, 0.2, and 2 mg/m^3^ MWNT-7 for 2 years as reported in Kasai et al. 2016 [14] are in shown in Table 1. There is an excellent correlation between lung burden and total exposure (exposure concentration x number of exposures) for both male and female rats (Figure 3). The correlation coefficient is 0.99 for both males and females (Table 2).

### 3.3. BALF and Abdominal and Thoracic Lavage Fluid Findings

Total protein (TP) in the BALF is an indicator of capillary permeability. The correlation between lung burden and TP in the BALF is shown in Figure 4. There was an excellent correlation between TP and lung burden. The correlation coefficient is 0.95 for males and 0.91 for females (Table 2). There was also an excellent correlation between total exposure (exposure concentration × number of exposures) and TP in the BALF for both sexes. The correlation coefficient is 0.97 for males and 0.99 for females (Table 2). 

Lactate dehydrogenase (LDH) in the BALF is an indicator of alveolar epithelial cell cytotoxicity. The correlation between lung burden and LDH in the BALF is shown in Figure 5. There was a good correlation between lung burden and LDH. The correlation coefficient is 0.71 for males and 0.75 for females (Table 2). A high correlation was also observed between total exposure and LDH in the BALF for both sexes. The correlation coefficient is 0.87 for males and 0.95 for females (Table 2). 

Alkaline phosphatase (ALP) in the BALF is an indicator of alveolar type II epithelial cell toxicity. An excellent correlation was observed between LDH and ALP in the BALF with correlation coefficients of 0.87 for males and 0.95 for females (Table 2). 

In the one-day and two-week exposure groups, one day after the end of the exposure period there was a decrease in macrophage counts in the BALF of MWNT-7 exposed rat study [6,7]. In the 2-week exposure study, this decrease was MWNT-7 dose dependent [7]. As shown in Figure 6, in the 1-day exposure study (5 mg/m^3^), no recovery in macrophage numbers was observed 7 days after the exposure period; however, at 28 days after the last exposure, macrophage counts in the 5 mg/m^3^ exposure group had recovered to approximately the same level as in the control group. In contrast, in the 2-week exposure groups, while the macrophage number recovered to about 85% of the controls at 4-weeks post exposure in males, in females there was very little recovery of macrophage number at 4-weeks post exposure [7]. This sex difference needs further study. The appearance of multinucleated macrophages phagocytosing MWNT-7s was a characteristic feature in these rats [7], and this may have affected macrophage number in the BALF.

The decrease in macrophages observed in the 1-day and 2-week studies was not observed at the end of the 13-week or 2-year studies. In the 13-week study there was almost no change in macrophage numbers in the BALF of male or female rats, and in the 2-year study there was a noticeable increase in macrophage counts in the BALF of female rats at the end of the study compared to the controls. Notably, macrophage number in the BALF at the end of the study increased with increasing exposure level. The 2-year BALF examination showed a dose-dependent increase in inflammatory cells such as neutrophils, lymphocytes, eosinophils, and macrophages in both males and females. This increase was especially striking in females at 2 mg/m^3^. Phagocytosis of MWNT-7 by macrophages was a common feature of all of the studies. In the 2-year study, many disintegrated macrophages were observed due to phagocytosis of MWNT-7 fibers. Photographs of macrophages in the BALF of the 2-year study are shown in Figure 7. It is likely that phagocytosis of MWNT-7 by alveolar macrophages had a chemotactic effect, resulting in the observed increase in macrophage number in the BALF. 

In the pleural lavage fluids in the 2-year carcinogenicity study numerous inflammatory cells were observed (Figure 8). Notably, mast cells, which were not observed in the BALF, and eosinophils, which were scarce in the BALF, were conspicuous. These changes were not observed in the hematological examination of rats exposed to MWNT-7 in the 2-week, 13-week, or 2-year studies. 

### 3.4. Lung Morphology and Pathological Examinations

In the 1-day study [6], histopathological examination of lung tissue revealed macrophages with phagocytosed MWNT-7 fibers immediately after the end of the 6 h exposure period and macrophages with phagocytosed MWNT-7 fibers and free long MWNT-7 fibers were observed in the alveolar space at 0, 1, 7, and 28 days after exposure. Furthermore, early-stage granulomatous lesions due to aggregation of alveolar macrophages with phagocytosed MWNT-7 fibers were observed on day 28 after the end of exposure [6]. 

In the 2-week study [7], granulomatous lesions were observed at the end of the exposure period in 1/5 male and 2/5 female rats exposed to 5 mg/m^3^ MWNT-7. At 4-weeks post exposure, granulomatous lesions increased to 4/5 males and 5/5 females. In addition to granulomatous lesions, a small amount of collagen fibers in the alveolar wall was observed at 4-weeks post exposure, suggesting early-stage fibrosis. 

In the 13-week study [8], granulomatous lesions were observed in the alveoli of both males and females exposed to 1 mg/m^3^ (8/10 males and 4/10 females) and 5 mg/m^3^ (10/10 males and 10/10 females). In addition, focal alveolar fibrosis was observed in nearly all males and females in these two exposure groups. In this study, granulomatous lesions were closely associated with focal fibrosis. 

In the 2-year carcinogenicity study [14], granulomatous lesions and fibrosis at the alveolar wall were observed in all three exposure groups (0.02, 0.2, and 2 mg/m^3^). Granulomatous lesion and focal fibrosis were found in 5/50 and 2/50 males exposed to 0.02 mg/m^3^, 42/50 and 43/50 males exposed to 0.2 mg/m^3^, and 50/50 and 48/50 males exposed to 2 mg/m^3^, respectively. Granulomatous lesions and focal fibrosis were found in 3/50 and 3/50 females exposed to 0.02 mg/m^3^, 45/50 and 44/50 females exposed to 0.2 mg/m^3^, and 50/50 and 49/50 females exposed to 2 mg/m^3^, respectively. Thus, similar to the 13-week study, the appearance of granulomatous lesions was closely associated with the development of fibrosis of the lungs. However, in contrast to the 13-week study, epithelial hyperplasia lesions such as bronchiolo–alveolar hyperplasia and alveolar hyperplasia in the lung were observed in the two-year study. The occurrence of bronchiolo–alveolar hyperplasia, which is considered a preneoplastic lesion, was observed in 2/50, 6/50, 13/50, and 22/50 males exposed to 0. 0.02, 0.2, and 2 mg/m^3^, respectively, and in 3/50, 3/50, 8/50, and 12/50 females exposed to 0. 0.02, 0.2, and 2 mg/m^3^, respectively. The occurrence of lung tumors, histopathologically diagnosed as adenomas and carcinomas, was similar to the occurrence of bronchiolo–alveolar hyperplasia. Lung tumors were found in 2/50, 2/50, 13/50, and 16/50 males exposed to 0. 0.02, 0.2, and 2 mg/m^3^, respectively, and in 3/50, 2/50, 4/50, and 11/50 females exposed to 0. 0.02, 0.2, and 2 mg/m^3^, respectively. The development of lung carcinomas, mainly bronchiolo–alveolar carcinomas, and combined adenomas and carcinomas was significantly increased in males exposed to 0.2 and 2 mg/m^3^ MWNT-7 and in females exposed to 2 mg/m^3^ MWNT-7 compared to the clean-air control group.

## 4. Discussion

### 4.1. Lung Burden

In the 1-day exposure to 5 mg/m^3^ MWNT-7 fibers, comparison of the lung burden immediately after the end of the 6 h exposure period and the lung burden the day after exposure indicated that approximately 50% of the MWNT-7s were cleared from the lungs by the next day. It is likely that the rapidly cleared MWNT-7 fibers were deposited in the ciliated airways up to the terminal bronchioles and were removed from the lung by the mucociliary escalator, and that the retained fibers were deposited beyond the ciliated airways in the alveolar region. Clearance of the fibers that were presumably deposited in the alveolar region was slow, with approximately 88% after 7 days, 81% after 28 days, and 78% after 56 days of the MWNT-7 fibers remaining in the lung based on next-day values. It is also notable that removal of fibers from the lung decreased over time. As reported in Section 3.2 above, MWNT-7 deposition per whole lung after 6 h (1 day) exposure to 5 mg/m^3^ was 8.7 ± 0.6 µg immediately after exposure, 4.6 ± 0.8 μg the next day, 4.1 ± 1.0 µg after 7 days, 3.7 ± 0.5 µg after 28 days, and 3.6 ± 0.6 μg after 56 days. Therefore, 0.5 µg of fibers were removed from the lung from day 1 after exposure to day 7; 0.4 µg of fibers were removed from the lung from day 7 after exposure to day 28; and 0.1 µg of fibers were removed from the lung from day 28 after exposure to day 56. This would result in deposition of fibers in the lung as a function of exposure concentration and duration.

The total amount of MWNT-7 fibers retained in the lungs of male rats exposed to 5 mg/m^3^ for 2 weeks was approximately 43.4 µg/lung one day after the end of the exposure period (Table 1). The 2-week lung burden was 9.4-fold greater than the lung burden of male rats exposed to 5 mg/m^3^ for one day (4.6 ± 0.8 μg/lung). In the 2-week study, there were 10 × 6 h exposure periods. Considering the clearance rate of fibers deposited in the alveolar region, discussed above, it is reasonable that the lung burden was about 10 times higher after 10 × 6 h exposure periods compared to a single 6 h exposure period. These results suggest that a general correlation exists between the total exposure dose (exposure concentration × number of exposures) and lung burden; as shown in Figure 3, there is a high correlation between lung burden and total exposure (exposure concentration × number of exposures) for both males and females. The correlation coefficient is 0.99 for both sexes (Table 2). 

In the 2-week study, in the 5 mg/m^3^ group, on day 1 after the end of the exposure period the lung burden was 43.4 µg/lung and on day 28 after the end of the exposure period the lung burden was 41.2 µg/lung [7]. This high fiber retention (95%) is higher than that seen in the 1-day study 28 days after the end of the exposure period (approximately 81%). However, this difference is in agreement with a continual low clearance of fibers from the lungs that decreases over time during the 2-week exposure period, i.e., the proportion of very slowly cleared fibers in the lungs after 2 weeks of exposure is higher than the proportion of very slowly cleared fibers in the lungs after 1-day exposure, resulting in higher overall retention of fibers in the lung after the end of the exposure period in the 2-week study compared to the 1-day study. 

Based on the percentage of MWNT-7 fibers remaining in the lungs on days 7, 28, and 56 post exposure compared to 1-day post exposure, the alveolar clearance rate of MWNT-7 is calculated to be Y = 4.5783 X^−0.061^ per lung for this study: Y is lung burden and X is days post exposure. Substituting 1, 7, 28, and 56 for X in this equation, Y (lung burden) is calculated to be 4.58, 4.07, 3.74, and 3.58, which approximates the actual measured values of 4.6, 4.1, 3.7, and 3.6 µg/lung (see Section 3.2). In this equation, when Y is 2.3 (clear half of the fibers out of the lung), the number of days is about 10^5^ days (about 274 years). While the equation shown here does not directly apply to the lung burdens in the 2-week, 13-week, or 2-year studies, separate equations need to be calculated for each individual exposure concentration and post-exposure lung burden, every study is expected to have similarly low clearance rates. Thus, the calculations that apply to the 1-day study indicate that except for extremely low concentration exposure levels the clearance half time of MWNT-7 out of the alveoli cannot be determined in rats with a life span of 2–3 years. Since the half life of macrophages is usually 60–90 days, the low clearance by macrophages of MWNT-7 fibers out of the alveoli in the 1-day study was not due to a short macrophage half-life.

Lung burden in our MWNT-7 inhalation studies can be briefly summarized as follows. During the initial 24 h after exposure approximately 50% of MWNT-7 is cleared from the lung, but little clearance of fibers from the lung occurs thereafter. This indicates that fibers deposited in the upper airways are rapidly cleared from the lung by the action of the mucus–cilia escalator function (ciliary motility), but removal of the fibers from the alveolar region is largely ineffective. Measurements of lung burden indicate that clearance of fibers deposited in the alveolar region has a half time of many years. This results in deposition of MWNT-7 in the lung in an exposure concentration and exposure time dependent manner.

### 4.2. Mechanisms of MWNT-7 Fiber Retention in the Lung

The theory that form and shape are closely related to the toxicity and carcinogenicity of fibrous materials was proposed by Stanton in 1972 [17] and Pott in 1974 [18]. The Stanton hypothesis states “Durable fibers, perhaps at the extreme ranges of dimension, cause cancer simply because they are fibers and irrespective of their physicochemical nature” [19,20]. In a seminal study using implantation of 72 types of fibrous materials onto the pleural surface of female rats, Stanton et al. reported that the pleural carcinogenicity of fibers depended on dimension and durability [21]. This study found that pleural sarcomas were induced by fibers with diameters up to 1.5 µm and lengths greater that 5 µm, and that the most carcinogenic fibers were those with diameters of 0.25 µm or less and lengths greater than 8 µm. Pott, mainly using intraperitoneal injection to investigate the carcinogenic potency of various fibers [18,22,23,24], developed the Stanton hypothesis further by quantitating the carcinogenic potency of a fiber based on its dimensions, with carcinogenic potential changing with length, diameter, and the length-diameter ratio [25,26] see also Schneider et al., 1985, [27] and Figure 11 in Adachi et al., 2001, [28]. As can be seen by their methodologies, the Stanton and Pott hypotheses were based on induction of mesothelioma by the test fibers. Examination of human lung tissue from mesothelioma patients also indicated that fiber dimension was an important carcinogenic factor; McDonald et al. examined lung tissue samples from 78 mesothelioma cases and concluded that relative mesothelioma risk was related to the concentration in the lung tissue samples of amphibole asbestos fibers more than 8 µm in length [29].

As noted above, the Stanton and Pott hypotheses and supporting evidence was based on fiber-induced mesothelioma in the plural and peritoneal cavities of rats. The first study to investigate the relationship of fiber dimension with lung carcinogenicity was by Davis et al., 1986, [30]. They exposed rats to long amosite, short amosite, and Union for International Cancer Control (UICC) amosite fibers by inhalation. Eleven lung tumors developed in the rats exposed to long fibers, no lung tumors developed in the rats exposed to short fibers, and two lung tumors developed in the rats exposed to UICC fibers. The short fiber preparation contained very few fibers greater than 5 µm in length while in the long fiber preparation over 11% of the fibers had lengths greater than 10 µm and 3% had lengths over 25 µm. These results suggest that the Stanton and Pott hypotheses, that one of the factors associated with the carcinogenicity of biopersistent fibers was fiber shape, was valid for fiber induced lung carcinogenesis as well as fiber induced mesothelioma.

More recently, the Stanton hypothesis has been applied to high aspect nanoparticles, which includes CNTs, and this has become known as the fiber pathogenicity paradigm [31,32]. The fiber pathogenicity paradigm associates fiber dimension with lung as well as pleural carcinogenicity. While the association of fiber dimension with retention in the pleural cavity and induction of mesothelioma is relatively straightforward [31], the association of fiber dimension with retention in the lung is more complex. The most obvious factor that would result in impaired macrophage motility and retention of fibers in the lung is frustrated phagocytosis of rigid fibers longer than approximately 15 µm [32]. In our 2-year carcinogenicity study, rats were exposed to fibers with a mean diameter of 92.9–98.2 nm and a mean length of 5.4–5.9 μm; the mean diameter and width of the fibers recovered from the lungs at the end of the 2-year exposure period were 95.5–109.6 nm and 5.8–5.9 μm [14]. This indicates that fibers that did not cause frustrated phagocytosis were retained in the lung. In support of this concept, macrophages in the state of frustrated phagocytosis were not observed in our 2-year study. While these findings do not eliminate frustrated phagocytosis as one of the mechanisms by which longer fibers were retained in the lung, they indicate that retention of MWNT-7 fibers in the lungs of exposed rats was not solely dependent on frustrated phagocytosis.

A second mechanism by which macrophage movement can be impaired is by exposure to excessively high levels of poorly soluble particles, resulting in particle overload in the lung. Impairment of particle clearance under these circumstances was proposed by Morrow to be due to the physical overloading of macrophages with a consequent loss of cell mobility [33]. Subsequent research has supported impaired macrophage mobility as a contributing factor to lung particle overload [34]. The IARC monograph on the evaluation of carcinogenic risks to humans Volume 81, Man-Made Vitreous Fibres, states “Particle overload occurs when high doses of poorly-soluble particles of low cytotoxicity are chronically deposited in the lung so that their daily rate of deposition exceeds the normal rate of macrophage-mediated clearance (ILSI, 2000)” [35]. Pauluhn argues that total particle volume is the most appropriate qualifier for particle overload [36]. Pauluhn reported that MWCNT overload and decreased MWCNT clearance began to occur at MWCNT volumes of approximately 1 µL per lung: the MWCNT used in the report are described in Pauluhn 2010 [37]. In our two-year inhalation study, individual fibers that were found in the lung at the end of the 2-year exposure had mean widths of 95.5–109.6 nm (calculated with a representative value of 102.55 nm), mean lengths of 5.8–5.9 µm (calculated with a representative value of 5.85 µm), and masses of approximately 1.1 × 10^−7^ µg/fiber; fiber mass can be calculated from the data presented in Table 4 in Kasai et al., 2016 [14]. The fibers collected from the lung immediately after a 6 h exposure to 5 mg/m^3^ MWNT-7 were 100.3 nm in diameter and 8.1 µm in length (Section 3.1). These fibers would have a mass of approximately 1.5 × 10^−7^ µg/fiber. The lung burden measured immediately after a 6 h exposure to 5 mg/m^3^ MWNT-7 was 8.7 ± 0.6 µg. This represents approximately 5.93 ± 0.4 × 10^7^ fibers/lung. 5.93 × 10^7^ fibers with diameters of 100.3 nm and lengths of 8.1 µm would have a volume of 3.8 × 10^15^ nm^3^, which is 0.0038 µL; since the MWNT-7 aerosols were composed of single well-dispersed fibers, fiber volume is the sum of the volumes of the individual cylinders, not the volumes of MWNT-7 agglomerates. This is far below the threshold at which lung burden is reported to affect clearance of particles from the lung. Even at the end of 2-years exposure to 2 mg/m^3^ MWNT-7, the 16.2 × 10^9^ fibers (mean width 95.5–109.6 nm and mean length 5.8–5.9 µm) found in the lungs of male rats would have a volume of only 0.78 × 10^18^ nm^3^ (0.78 µL), which is at the lower limit of lung burden affecting particle clearance [36]. Therefore, retention of MWNT-7 in the lungs of exposed rats was not due to particle overload in the lung. In addition, Pauluhn reported that the onset of delayed clearance occurred at exposure levels of 5 µL particles/m^3^. For the aerosolized MWNT-7 fibers generated in the 2-year inhalation study, this would be approximately 12 mg/m^3^, a level significantly higher than was used in the 2-year inhalation study. Finally, the cocoon-like masses of fibers that were found in macrophages in the 2-year inhalation study (Section 3.1) were found in both male and female rats exposed to all three levels of MWNT-7, 0.02, 0.2, and 2 mg/m^3^, indicating that they were not the result of exposure to excessively high levels of MWNT-7 fibers. Thus, retention of MWNT-7 in the lungs of exposed rats was not the result of particle overload.

A third mechanism that results in retention of particles in the lung is particle-induced alveolar macrophage cytotoxicity. In the 2-year study, biochemical analysis of the BALF indicated that there was no change in LDH or ALP levels in the 0.02 mg/m^3^ male or female groups compared to the controls. This indicates that there was no difference in alveolar epithelial cell cytotoxicity or type II epithelial cell toxicity in the 0.02 mg/m^3^ groups compared to the controls, and that there was no measurable tissue damage caused by macrophage cytotoxicity. In addition, we observed MWNT-7 fibers wound around each other in digested alveolar macrophages, forming large dense cocoon-like shaped masses 10–20 μm in diameter. Importantly, these masses were also found in macrophages in the lungs of rats exposed to the lowest concentration of MWNT-7, 0.02 mg/m^3^, in the 2-year study, and they were observed in the lungs in an MWNT-7 exposure concentration-dependent manner. Importantly, in our MWNT-7 inhalation studies the animals were exposed to well-dispersed single fibers. The formation of cocoon-like masses of fibers in macrophages, especially in the lowest exposure group of 0.02 mg/m^3^, argues that the formation of these fiber masses occurred over time. Thus, the presence of these masses coupled with the lack of measurable tissue damage strongly suggests that individual MWNT-7 fibers were not cytotoxic to macrophages.

However, in contrast to the argument presented above, we observed fibers associated with burst macrophages in the BALF (Figure 7). To explain these apparently contradictory findings, we propose a two-tiered approach to fiber-associated cytotoxicity: (i) particle-associated cytotoxicity that acts immediately upon interaction with the macrophage and acts independently of cytotoxic mediators generated by the macrophage and (ii) cytotoxicity associated with chronic production of reactive oxygen species and other cytotoxins by the macrophage. Particle-associated cytotoxicity would immediately damage the macrophage and inhibit macrophage motility, while the second type of cytotoxicity would cause damage to the macrophage over time, allowing initial movement of the macrophage out of the alveolar region. Overall, our data support the conclusion that phagocytosis of MWNT-7 fibers impair macrophage motility via a mechanism other than particle-associated cytotoxicity.

Currently, as discussed above, there are three mechanisms that are accepted as being associated with retention of carcinogenic levels of single (non-agglomerated) biologically durable fibers in the lungs of exposed rats: (1) frustrated phagocytosis, (2) lung particle overload, and (3) macrophage cytotoxicity. However, MWNT-7 fibers were not retained in the lung due to particle overload, and retention of MWNT-7 fibers in the lung did not depend on frustrated phagocytosis or macrophage cytotoxicity. This suggests that a fourth factor may be associated with impairment of macrophage movement into the ciliated airways and fiber retention in the lung. One possible mechanism is production of a fiber-induced “stop” signal by the macrophage that caused the macrophage to adhere strongly to the alveolar epithelium and not move into the ciliated region of the lung. Tumor necrosis factor (TNF), alpha, and interferon (IFN) gamma can negatively affect alveolar macrophage motility, functioning as “stop” signals [38]. Both of these cytokines activate p38 mitogen-activated protein (MAP) kinase, and consequently other factors that activate MAP kinase signaling, such as interleukin 1 (IL1) beta, also potentially function as stop signals. Notably, Francis et al. report that one time nose-only inhalation exposure of Wistar rats to MWCNT resulted in increased levels of TNF-alpha in the BALF [39]. Such a mechanism would have a length component as increased length would increase interaction with the phagocytic macrophage, thereby increasing production by the macrophage of mediators such as TNF-alpha and IFN-gamma.

Production of a “stop” signal agrees with the results of the one-day and two-week studies in which there was a decrease in macrophage counts in the BALF of MWNT-7 exposed rats, and this decrease was MWNT-7 dose-dependent. In this model of fiber-induced inhibition of macrophage movement, recovery of macrophage numbers in the BALF would be a consequence of inflammatory signaling by the fiber containing macrophages and chemotaxis of fiber-free macrophages into the alveoli. This model is also in agreement with the generation of the cocoon-like masses of fibers in macrophages observed in the 2-year study. Formation of these masses in macrophages exposed to the lowest concentration of MWNT-7, 0.02 mg/m^3^, suggests that macrophages which had phagocytosed only a few fibers, or possibly even a single fiber, remained in the alveoli where they subsequently phagocytosed additional fibers. Being able to phagocytose additional fibers indicates that these macrophages were biologically intact, and that while phagocytosis of the initial fiber(s) impaired macrophage motility, these fibers were not themselves cytotoxic to the macrophage. This possible fourth fiber associated factor requires additional investigation as our inhalation studies were not designed to investigate such a phenomenon.

Our proposal that fiber retention and fiber induced lung carcinogenesis can occur in the absence of frustrated phagocytosis agrees with a review of asbestos fibers and lung cancer that concluded that in humans, exposure to fibers longer than 5 µm, a length insufficient to result in frustrated phagocytosis, was associated with higher rates of lung cancer [40]. However, these authors are also careful to note that the pathogenicity of shorter asbestos fibers cannot be ruled out, especially in high-exposure situations. This also agrees with our definition of the possible fourth mechanism of fiber retention in the lung as phagocytosis of multiple shorter fibers would potentially have a similar effect as the phagocytosis of fewer longer fibers. This also agrees with the original Pott hypothesis which considers length, diameter, and length–diameter ratio, in calculating the carcinogenic potency of a fiber, suggesting that within certain limits a larger number of shorter fibers are as carcinogenic as a lower number of longer fibers [27]. The fourth mechanism of fiber retention proposed above also suggests that exposure dose is a critical distinction between the carcinogenicity of shorter and longer fibers. In low-exposure situations, with few fibers being phagocytosed by alveolar macrophages, short fibers would have relatively low fiber–macrophage interaction, resulting in low impairment of macrophage motility, and consequently would be cleared from the lung. In contrast, even in low-exposure situations, longer fibers would be retained in the lung, as seen in our 2-year study. In addition, our premise posits that longer fibers have higher activity in the lung compared to shorter fibers. These points are in agreement with the review by Boulanger et al. noted above that higher rates of lung cancer in humans were associated with longer asbestos fibers, but that lung cancer risk was also associated with shorter asbestos fibers [40].

### 4.3. Pathogenicity of MWNT-7 Fibers

We have reviewed the carcinogenicity of MWCNT in experimental animals, focusing on MWNT-7 and concluded that the inability of macrophages to degrade phagocytosed MWCNT fibers is closely related to the pulmonary carcinogenesis of MWCNTs [41]. In addition, the lung carcinogenicity induced by inhalation exposure to MWNT-7 (a fibrous, solid substance) in rats is dose related. Thus, retention of fibers in the lung is not sufficient to induce carcinogenesis, but in addition to this factor, the amount of fiber retained in the lungs and the inability of macrophages to degrade these fibers are crucial factors in fiber-mediated lung carcinogenesis.

MWCNTs are reported to have genotoxic activity [42,43,44,45,46,47,48]; however, MWNT-7 in particular is thought to induce genotoxicity through secondary mechanisms rather than as a direct DNA mutagen [42,43,44,45]. Macrophages produce reactive oxygen species (ROS) during the phagocytic response to MWCNT exposure [45,49], and ROS production by these macrophages is considered to be closely related to bronchiolo–alveolar hyperplasia and tumor development [41,43,45]. However, while production of ROS by macrophages could possibly cause mutations in adjoining epithelial cells, ROS are extremely reactive molecules and may not be able to pass through the barriers separating their origin in the macrophage phagosome and the DNA in the nucleus of the adjoining epithelial cells. For example, the diffusion-controlled reaction kinetics of hydroxyl radicals indicate that the target must be within approximately 10 Å of the site of hydroxyl radical generation [50,51]. Another possibility, given the association of inflammation with tumorigenesis, is that MWCNT-induced genotoxicity may not be a direct reaction between macrophage generated ROS and DNA but may result from mutations during repair of DNA damage caused by chronic inflammation, oxidative stress, cytotoxicity, and tissue damage induced by these factors. Importantly, Hojo et al. report that transcriptomic analysis of lung tissues from a 2-year study of rats administered MWNT-7 by intratracheal instillation supports a scenario of inflammation-induced carcinogenesis [52,53].

As discussed above, we propose a two-tiered approach to fiber-associated macrophage cytotoxicity: (i) particle-associated cytotoxicity that acts immediately upon association of the particle with the macrophage and (ii) cytotoxicity associated with generation by the macrophage of reactive oxygen species and other cytotoxins. Phagocytosis of MWNT-7 fibers by macrophages was confirmed by examination of the BALF in all of the inhalation studies from the 1-day exposure study to the 2-year carcinogenicity study, and damaged macrophages were observed in the 2-week, 13-week, and 2-year studies, but not in the 1-day study. This agrees with our conclusion that MWNT-7 fibers did not exhibit particle-associated cytotoxicity and that initial impairment of macrophage movement was caused by a non-cytotoxic mechanism, possibly generation of a stop signal. These data also indicate that exposure to MWNT-7 can cause alveolar macrophage damage via macrophage generated cytotoxins, and that this in turn promotes inflammation. Release of cytotoxic molecules from the damaged macrophages could also directly damage alveolar epithelial cells. TP in the BALF is an indicator of permeability of the alveolar air–blood barrier, and ALP and LDH levels in the BALF are indicators of alveolar cytotoxicity. As shown in Figure 4, there is an excellent correlation between lung burden and TP, and there is also a good correlation between lung burden and LDH levels in the BALF (Figure 5). These data indicate that pulmonary toxicity is proportional to lung burden. It is also clear that during inhalation exposures to MWNT-7 over several weeks to 2 years, prolonged toxicity reactions in the lung to the respired fibers occur during the period of exposure.

The initiating events of fiber-induced tumorigenesis in the pleural/peritoneal cavity and the lung are distinct. In the pleural/peritoneal cavity fiber-induced damage of mesothelial cells [10,54,55] and release of factors such as HMGB1 from the damaged cells [55,56,57,58] are critical initiating events of fiber-induced carcinogenesis while in the lung activation of macrophages and chronic production of cytotoxic molecules are key initiating events [41,43,59]. As noted above, damaged macrophages were observed in the BALF in the 2-week, 13-week, and 2-year studies (see Figure 4), indicating chronic production of cytotoxic molecules as the macrophages attempt to destroy the phagocytosed MWNT-7 fibers. The consequent tissue damage and inflammatory signaling resulted in neutrophil infiltration into the lung. Macrophage numbers in the BALF were also increased in the 13-week and 2-year studies, suggesting a response to inflammatory signaling. As discussed above, macrophage numbers in the BALF were reduced after the 1-day (6-h) exposure to MWNT-7, which was likely associated with fiber-mediated decreased macrophage motility and consequent retention of macrophages in the alveoli, and macrophage numbers in the BALF were also reduced after the 2-week exposure to MWNT-7, which was likely associated with increased retention of macrophages in the alveoli coupled with macrophage cytotoxicity consequent to chronic generation of cytotoxic molecules. These data can be summarized as follows. In line with their normal response to inhaled dusts, macrophages phagocytosed the MWNT-7 fibers. However, the motility of these macrophages was suppressed by a non-cytotoxic mechanism, and they remained in the alveolar region of the lung. Inflammatory signaling by these macrophages and generation of ROS and other cytotoxic molecules in response to the phagocytosed fibers resulted in inflammation in the lung and tissue damage. Chronic generation of cytotoxic molecules damaged the macrophages, resulting in additional release of tissue damaging agents. Inflammatory signaling by intact macrophages and signals from the damaged macrophages resulted in increased infiltration of neutrophils and macrophages into the lung. Examination of the BALF indicated that increased neutrophil infiltration into the lung was present immediately after 6-h exposure to 5 mg/m^3^ MWNT-7 and tissue damage was apparent 1 day after exposure to 5 mg/m^3^ MWNT-7, and neutrophil infiltration and tissue damage was still taking place at the end of 2-years exposure to MWNT-7. Clearly, two years of tissue damage is not sustainable, indicating that tissue repair was also occurring during this period. As noted above, one possible cause of MWNT-7 secondary genotoxicity is mutations occurring during repair of damaged tissue.

Two mechanisms are involved in retention of MWNT-7 fibers in the lung. The first mechanism, discussed above, is the initial interaction of fibers with phagocytosing macrophages resulting in decreased macrophage motility and consequent low fiber clearance from the lung. The other mechanism by which MWNT-7 fibers are retained in the lung is the result of macrophages with phagocytosed fibers forming foreign-body granulomatous lesions. This clears the respiratory area and decreases interaction between the foreign-body and lung tissue. Early-stage granulomatous lesions were observed in the lungs of one of five male rats 28 days after exposure to 5 mg/m^3^ MWNT-7 for 6 h (1 day) [6]. In the 2-week study, early stage granulomatous lesions were observed in the lungs of one of five male rats and two of five female rats at the end of the 2-week exposure period and in the lungs of four of five male rats and five of five female rats at the end of the 4-week post-exposure period [7]. Figure 9A shows the incidence of granulomatous lesions and lung burden for exposures ranging from one day to 2 years. As expected, there is a good correlation between the incidence of granulomatous lesions and lung burden. A 50% incidence of granulomatous lesions occurred at a lung burden of 50 µg. However, in the 2-week study granulomatous lesions increased during the 4-week post-exposure period, despite an unchanged lung burden. Therefore, a simple lung burden approach does not work. A duration component must be added. We re-examined the occurrence of granulomatous lesions in the one-day and two-week studies, including results from satellite groups not included in the original publication. Assuming no change in lung burden during the 28 days from the end of exposure to necropsy, lung burden at the end of each exposure period was multiplied by 28 days and the resultant area under the curve (AUC) values were plotted on the x-axis (Figure 9B). Using the AUC of lung burden and duration, the 50% incidence of the development of granulomatous lesions was 400 µg. Lesions that developed over time in our inhalation studies are better explained by the concept of the AUC of lung burden and duration.

In the 2-week study, a small amount of collagen fibers in the alveolar wall were observed at 4-weeks post exposure, suggesting early-stage fibrosis occurred with a similar lung burden and duration dependency as granulomatous lesions. In the 13-week and 2-year studies, granulomatous lesions and localized fibrosis of the alveolar wall were largely coincident, and the development of fibrosis was closely associated with the appearance of granulomatous lesions. These observations suggest a continuum between granulomatous lesions and fibrotic lesions, as is known to occur in various human conditions [60,61].

Similarly to the above discussion of the development of granulomatous lesions, when considering a correlation between lung burden and lung tumor incidence a duration component must be added. We constructed an AUC for lung burden and duration using the following provisions: (1) lung burden measured at the end of the exposure period was used, (2) deposition is zero on exposure day 0, and (3) increasing lung burden per day is assumed to be linear until the end of the exposure period. Figure 10 shows the AUC for lung burden and tumor incidence for the 13-week and 2-year studies. The AUC for the 5 mg/m^3^ group in the 13-week study is 16,241 µg for males and 10,841 µg for females. Using these calculations as references indicates that tumors that form in rats with AUCs for lung burden less than 16,241 µg in males and 10,841 µg in females cannot be considered to be caused by exposure to the test fibers. Therefore, it is reasonable to conclude that the lung tumors observed in the male and female groups exposed to 0.02 mg/m^3^ were spontaneous, indicating a dosage component to fiber-induced carcinogenesis.

This dosage component is in agreement with the random nature of the mutations resulting from fiber-induced chronic inflammation and consequent cycles of tissue damage and tissue repair. A lower fiber load will result in lower levels of inflammation and less tissue damage and tissue repair and fewer DNA mutations. Consequently, at a lower fiber load it will take longer on average to generate a full set of carcinogenic mutations.

We propose the following pathway for MWNT-7 induced lung cancer. Inhaled fibers that are deposited in the ciliated airways are quickly cleared from the lung while fibers deposited beyond the ciliated airways require phagocytosis and movement into the ciliated airways by alveolar macrophages in order to be cleared. Interaction with alveolar macrophages results in impaired macrophage motility via a non-cytotoxic mechanism. Interaction of the fiber with the macrophage also results in inflammatory signaling and generation of cytotoxic molecules to destroy the “invading” fiber. The inability of the macrophage to destroy the fiber results in continuous generation of cytotoxic molecules and inflammatory signaling. This results in macrophage cytotoxicity, tissue damage, and inflammation. Fibers released from damaged macrophages are phagocytosed by other macrophages, resulting in continued inflammation and tissue damage. Tissue damage is followed by tissue repair, and tissue repair, especially in the presence of cytotoxic molecules, results in a low level of DNA mutations. Chronic inflammation and the consequent cycles of tissue damage and repair can eventually result in the acquisition by a cell of a full set of mutations required for cell transformation. This pathway suggests that any inhaled carbon nanotube that induces impaired macrophage motility is a potential carcinogen.

### 4.4. Comparison with Intratracheal Instillation

Long-term 2-year studies of MWNT-7 and MWCNT-N have been carried out using administration by intratracheal instillation [52,62,63,64]. These studies are summarized in Figure 9 by Hojo et al., 2022 [52]. The lung burden/tumor incidence of the inhalation studies cannot be directly compared with that of the intratracheal and intrapulmonary spraying (TIPS) studies: in the study by Abdelgied et al., three untreated controls developed lung cancer making an incidence of 43% in the MWCNT-7 administered groups not significantly increased over controls [62]; Suzui et al. used MWCNT-N rather than MWNT-7 [64]; and in the study by Numano et al., 18 of the MWNT-7 administered rats died from malignant mesothelioma of pleural cavity [63]. However, it is possible to directly compare the lung burden/tumor incidence of the inhalation studies with that of the intratracheal administration study by Hojo et al. [52]. Using the curves generated by Hojo et al. the AUCs for the lung burdens (and tumor incidence) in the 0.2 mg/m^3^ and 2 mg/m^3^ MWNT-7 inhalation exposed groups are 7.5 mg (26%) and 90 mg (32%), respectively, and the AUCs for the lung burdens (and tumor incidence) in the intratracheal administration groups are a little less than 45 mg (10%) and 180 mg (39%), respectively. These data indicate that tumor development was observed at a lower lung burden when MWNT-7 was administered via the inhalation exposure route compared to intratracheal administration. The fact that tumor development was observed in the inhalation study even though the AUC for lung burden was smaller than that for intratracheal administration may be due to differences in the length and width of the test substance, the frequency of administration, and the fibers per dose due to the administration method. Inhalation exposure is characterized by the fact that small amounts of individual MWCNT fibers enter the alveoli through breathing, and although some of the MWCNTs that enter the alveoli are removed, many gradually accumulate in the macrophages. On the other hand, in intratracheal administration, a large amount of MWCNTs is sprayed into the lung during a single inhalation, and consequently, dispersal in the lung will differ from the dispersal pattern in inhalation studies. Necropsy findings of the lung support this hypothesis. We observed the lung coloration at the terminal necropsy as follows: “At the terminal necropsy, multiple grayish, white areas and nodules were found in the lungs of a large number of male and female rats exposed to 2 mg/m^3^ MWNT-7. The color tone of the lung surface was darkened in line with exposure concentration” [14]. Hojo et al. [52] described lung coloration of rats sacrificed at weeks 26 and 52 as “gray to black colored MWCNT depositions. The degree of darkness at the lung surface increased with the dose and time”. Comparing the terminal necropsy photographs attached to both papers, the 2 mg/m^3^ group in our 2-year paper appears uniformly darker in color, whereas the necropsy photograph of the high-dose group in Hojo et al. has areas that appear normal in color. This difference in the coloration of the lung surface indicates that MWNT-7 fibers reach the subpleural region of the lung diffusely in inhalation exposure, while the somewhat localized observation of intratracheal administration suggests less deposition in the subpleural region. Another likely factor pertaining to the difference in the carcinogenicity of inhaled fibers versus fibers administered by intratracheal instillation is altered kinetics in the formation of foreign-body granulomatous lesions in the lung. Bolus administration of fibers likely results in rapid formation of granulomatous lesions. Sequestration of fibers in granulomas reduces fiber interaction with macrophages, thereby decreasing the inflammatory signaling, macrophage cytotoxicity, and tissue damage generated on a per fiber basis. In contrast, each inhaled fiber deposited beyond the ciliated airways has the potential to interact with alveolar macrophages and induce the generation of inflammatory signaling, macrophage cytotoxicity, and tissue damage. Thus, on a per fiber basis, the dose of alveolar macrophage reactive fibers administered by instillation is lower than that administered by inhalation.

## 5. Conclusions

To evaluate the toxicity of multi-walled carbon nanotubes in rats, we have developed a unique MWNT-7 generator based on dry aerosolization using the aerodynamic cyclone principle. Inhalation exposure studies using this device lasting 1 day, 2 weeks, 13 weeks, and 2 years were conducted in accordance with OECD test guidelines and GLP principles, and the results of these studies have been published.

Generation of well-dispersed single fibers avoided the confounding effects of fiber agglomeration in the evaluation of the results of these inhalation studies. After 6 h exposure to a relatively high level (5 mg/m^3^) of MWNT-7 fibers, approximately 50% of the deposited MWNT-7 was cleared during the initial 24 h post exposure. These fibers were likely deposited in the ciliated airways and cleared by mucociliary transport. The fibers that were deposited beyond the ciliated airways had a very low clearance rate. The data from our studies strongly suggest that a mechanism distinct from frustrated phagocytosis and macrophage cytotoxicity was responsible for initially inhibiting the motility of macrophages which had phagocytosed these relatively short fibers. This inhibitory mechanism is a key factor in the carcinogenicity of MWNT-7 observed in our 2-year study. Inhibition of macrophage movement out of the alveolar region resulted in phagocytosis of additional fibers deposited in the alveoli which further inhibited movement of these macrophages out of the alveolar region. Consequently, the inability of these macrophages to move into the ciliated airways coupled with their inability to degrade the fibers resulted in chronic generation of inflammatory signaling and generation of reactive oxygen species and other cytotoxic molecules, resulting in lethal damage to many of the macrophages and damage to the lung tissue. Over time, the chronic inflammatory signaling, macrophage cytotoxicity, and the consequent cycles of tissue damage and tissue repair resulted in the accumulation of DNA mutations and eventually in the development of lung adenomas and carcinomas. The carcinogenicity incidence was consistent with the AUC of fiber lung burden and duration.

In addition, macrophages with phagocytosed MWNT-7 fibers that were not destroyed by the cytotoxic molecules they generated in response to the phagocytosed fibers formed foreign-body granulomatous lesions. Granulomatous lesions were associated with the development of fibrosis. While our studies were designed to investigate MWNT-7 induced carcinogenesis and not fibrosis, pulmonary fibrosis can be a serious lung disease. Granulomatous lesions also developed in consanguinity with the AUC of fiber lung burden and duration.

## 6. Patents

This paper summarizes toxicity and carcinogenicity studies conducted on behalf of the Japanese Ministry of Health, Labor and Welfare (MHLW).

## Figures and Tables

**Figure 1 nanomaterials-13-02598-f001:**
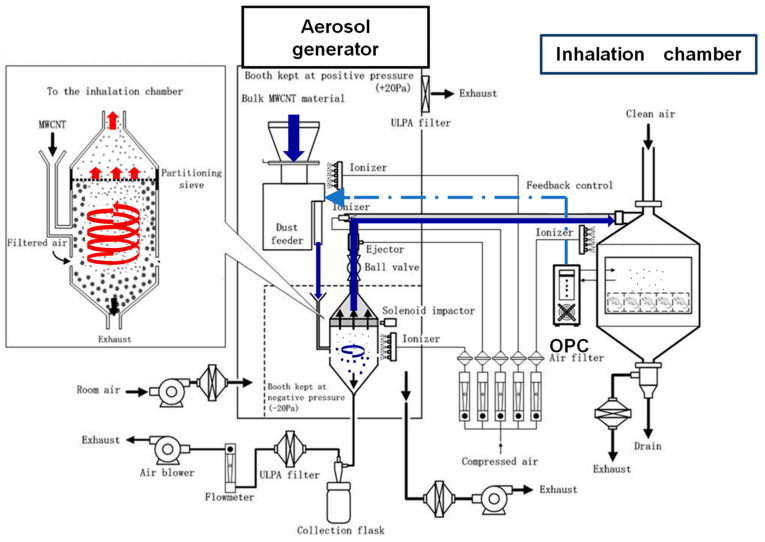
Layout of the aerosol generator (cyclone sieve method) and inhalation exposure system for MWCNTs. The red arrows in the left panel indicate aerosolization of MWNT-7 by the upward spiraling airstream. Dark blue arrows indicate the flow of MWNT-7 from the aerosol generator to the inhalation chamber. First, MWNT-7 (bulk MWNT-7) is fed into the dust feeder. The dust feeder transports MWNT-7 into the sieving unit (left panel). In the sieving unit, clean air is aspirated from nine diagonally opened slits using the ejector air as the driving force, so that an upward spiraling airstream is continuously generated in the cylindrical container. MWNT-7 is dispersed and aerosolized by the high-speed spiraling air. Light MWNT-7 particles are carried to the top of the cylindrical container, where the partitioning sieve is located. Only fine MWNT-7 fibers that pass through the sieve can be delivered to the whole-body inhalation chamber. Several ionizers are deployed to avoid agglomeration. The device has a feedback system (light blue dashed arrow) to control the supply of MWNT-7 into the sieving unit by the dust feeder, maintaining a constant aerosol concentration in the inhalation exposure chamber.

**Figure 2 nanomaterials-13-02598-f002:**
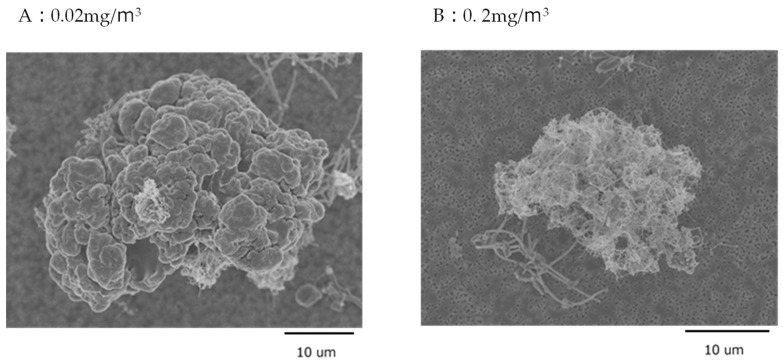
SEM images of macrophage aggregates observed in the lungs of rats exposed to MWNT-7 for 2 years. Left panel (**A**): Macrophage aggregates present in the BALF. Many macrophages aggregated to form a large mass. Right panel (**B**): Complete digestion of a macrophage aggregate showing MWNT-7 fibers from the individual macrophages form small cocoon-like masses which remain aggregated after digestion of the macrophage aggregate.

**Figure 3 nanomaterials-13-02598-f003:**
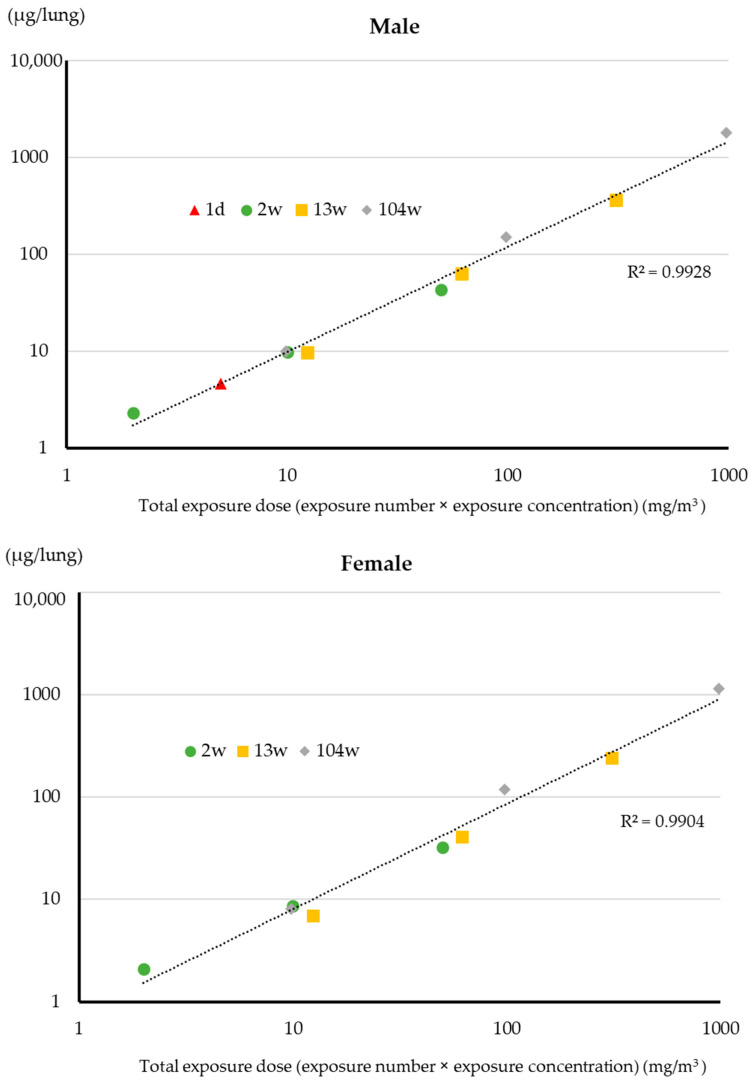
Lung burden versus total exposure dose (exposure concentration × number of exposures) for male and female rats exposed to three concentrations of MWNT-7 in 2-week, 13-week, and 2-year studies is shown. Male results include the one-day exposure to 5 mg/m^3^.

**Figure 4 nanomaterials-13-02598-f004:**
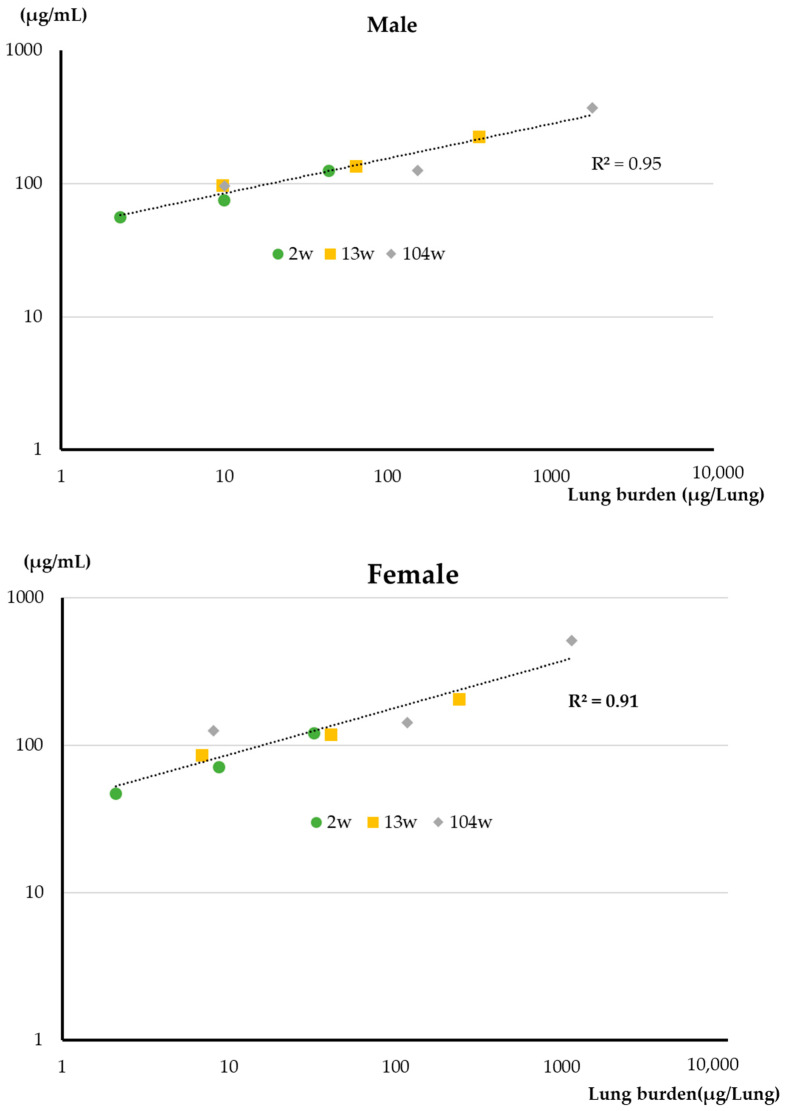
Relationship between lung burden and total protein in the bronchoalveolar lavage fluid (BALF) in male and female rats exposed to three concentrations of MWNT-7 in the 2-week, 13-week, and 2-year studies.

**Figure 5 nanomaterials-13-02598-f005:**
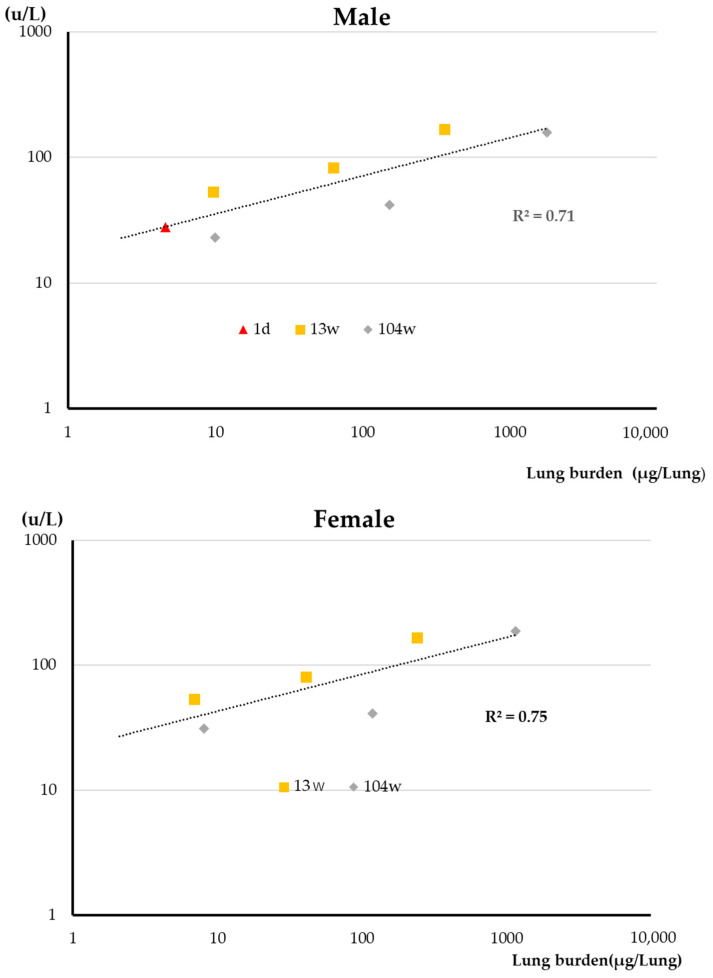
Relationship between lung burden and LDH in the bronchoalveolar lavage fluid (BALF) in male and female rats exposed to three concentrations of MWNT-7 in the 13-week and 2-year studies. Male results include one-day exposure to 5 mg/m^3^.

**Figure 6 nanomaterials-13-02598-f006:**
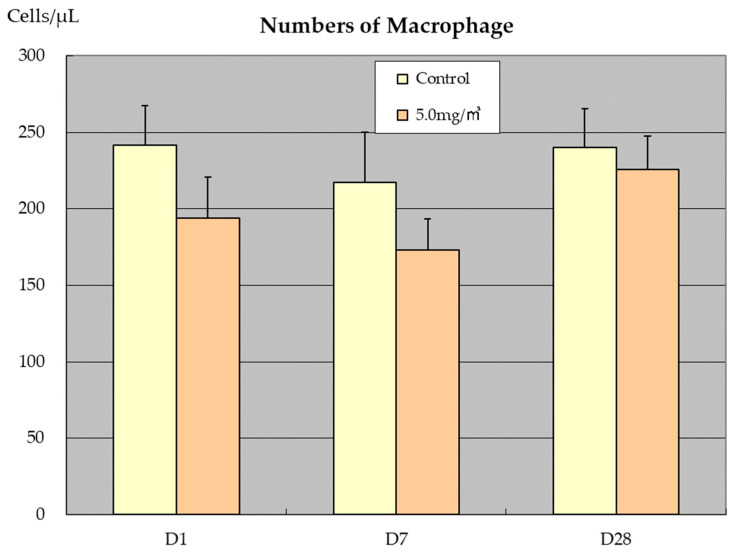
Alveolar macrophage counts in the BALF of rats 1, 7, and 28 days after exposure to 5 mg/m^3^ MWNT-7 for 1 day (6 h).

**Figure 7 nanomaterials-13-02598-f007:**
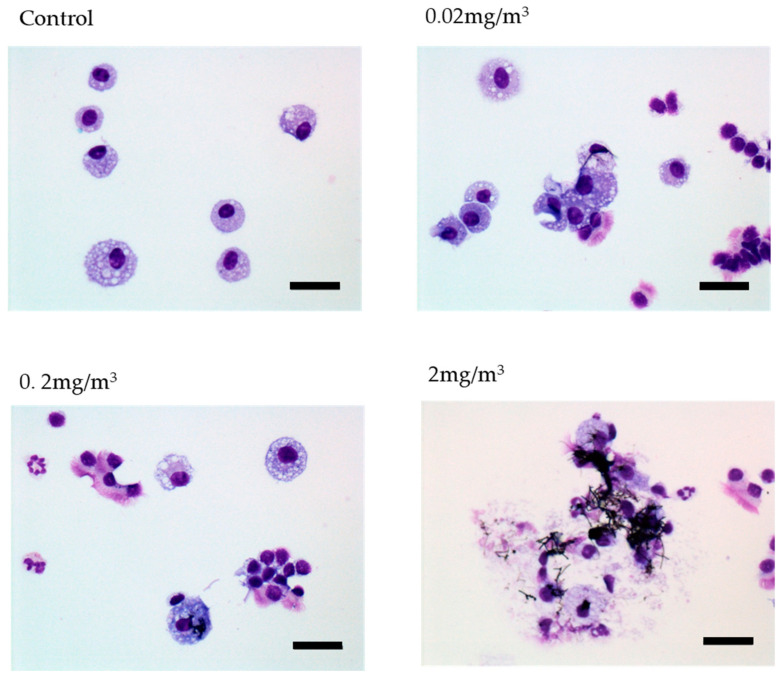
Alveolar macrophages (at the end of 2 years of exposure) in the BALF of rats exposed to MWNT-7 at concentrations of 0, 0.02, 0.2, and 2 mg/m^3^. Multinucleated macrophages and alveolar macrophages with cytoplasm loss due to phagocytosis by MWNT-7 are observed. Cells stained with May–Grünwald–Giemsa stain. Bars indicate 30 μm.

**Figure 8 nanomaterials-13-02598-f008:**
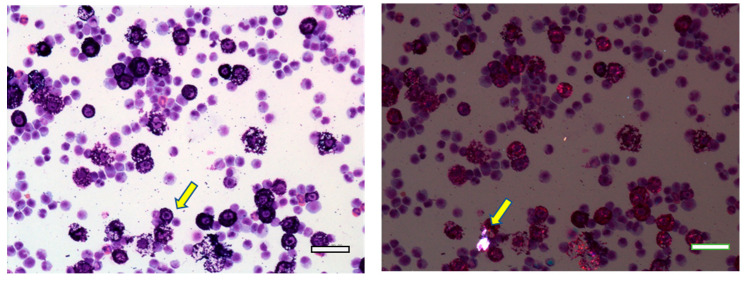
Pleural lavage fluid of a female rat exposed to 2 mg/m^3^ MWNT-7 for 6 h/day for 2 years. The left panel is stained with May–Grünwald–Giemsa stain. The right panel is a phase-contrast polarized light micrograph of the left panel. Arrows indicate mast cells. The bright white object just under the arrow in the right panel is a cluster of MWNT-7 fibers. Bar indicates 90 µm.

**Figure 9 nanomaterials-13-02598-f009:**
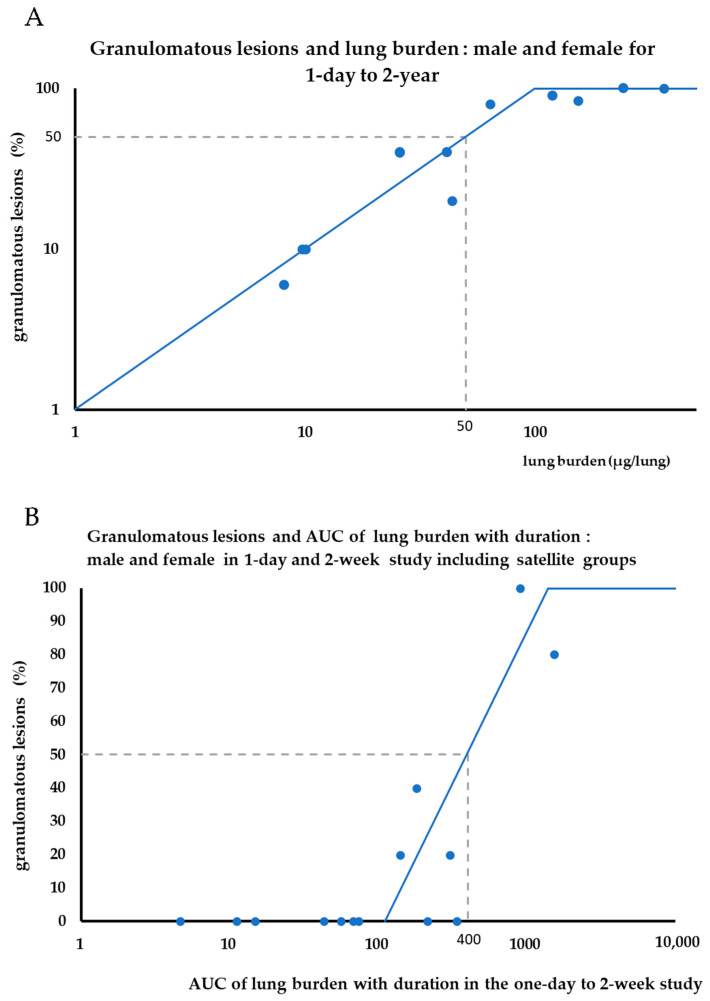
Lung burden and incidence of granulomatous lesions in male and female rats observed in the 1-day, 2-week, 13-week, and 2-year studies. (**A**) The lung burden in both males and females is plotted on the x-axis, and the granulomatous lesion incidence rate (%) is plotted on the y-axis. The dotted line connects the lung burden values that represent the 50% value for the occurrence of granulomatous lesions. (**B**) The AUC values of lung burden and the incidence of granulomatous lesions observed in male and female rats in the 1-day and 2-week studies, including data from satellite groups not included in the original publication. The AUC values of lung burden for the 1-day and 2-week studies are plotted on the x-axis, and the granulomatous lesion incidence rate (%) is plotted on the y-axis. The AUC of lung burden values that represent the 50% value of the incidence of granulomatous lesions is indicated by the dotted line. The dotted line connects the lung burden values that represent the 50% value for the occurrence of granulomatous lesions.

**Figure 10 nanomaterials-13-02598-f010:**
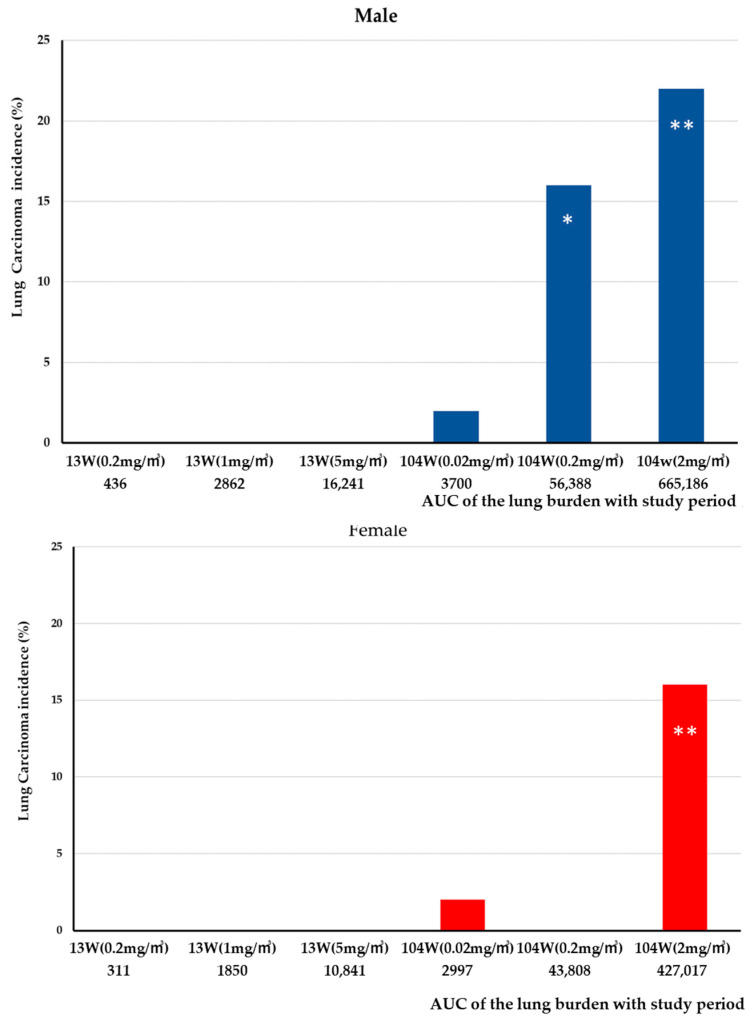
AUC values of lung burden and lung Carcinoma incidence (%) in the 13-week and 2-year studies. The exposure concentrations (mg/m^3^) and the AUC values of lung burden in the 13-week and 2-year studies are shown on the x-axis, and carcinoma incidence (%) is shown on the y-axis. Significant difference; *: *p* ≤ 0.05 **: *p* ≤ 0.01 using Fisher’s exact test.

**Table 1 nanomaterials-13-02598-t001:** Lung burden analysis.

**Male**				
**Study period**	**Exposure Conc. (mg/m^3^)**	**Exposure Times (Number of days)**	**Concentration × number of exposures**	**Lung burden (μg)/Whole Lung (mean ± SD)**
One-day	5	1	5	4.6 ± 0.8
	0.2	10	2	2.3 ± 0.4
2-week	1	10	10	9.9 ± 1.2
	5	10	50	43.4 ± 9.3
	0.2	62	12.4	9.7 ± 0.8
13-week	1	62	62	63.6 ± 3.9
	5	62	310	360.9 ± 51.9
104-week (2-year)	0.02	492	9.84	10.0 ± 2.6
	0.2	492	98.4	152.4 ± 19.4
	2	492	984	1797.8 ± 146.0
**Female**				
**Study period**	**Exposure Conc. (mg/m^3^)**	**Exposure Times (Number of days)**	**Concentration × number of exposures**	**Lung burden (μg)/Whole Lung (mean ± SD)**
	0.2	10	2	2.2 ± 0.4
2-week	1	10	10	8.7 ± 2.3
	5	10	50	32.5 ± 9.7
	0.2	62	12.4	6.9 ± 0.7
13-week	1	62	62	41.1 ± 6.6
	5	62	310	240.9 ± 41.1
104-week (2-year)	0.02	492	9.84	8.1 ± 1.8
	0.2	492	98.4	118.4 ± 13.4
	2	492	984	1154.1 ± 77.6

Lung burden analysis was performed on the day after the final exposure.

**Table 2 nanomaterials-13-02598-t002:** Correlation coefficients.

**Males**			
	**Lung Burden**	**Total Protein**	**LDH**
Total Exposure	0.99	0.97	0.93
Lung Burden		0.95	0.71
ALP			0.87
**Females**			
	**Lung Burden**	**Total Protein**	**LDH**
Total Exposure	0.99	0.99	0.81
Lung Burden		0.91	0.75
ALP			0.95

Note: Total Exposure is the value of exposure concentration x number of exposures

## Data Availability

Not applicable.

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
