# Peer review of "Exposure of Rats to Multi-Walled Carbon Nanotubes: Correlation of Inhalation Exposure to Lung Burden, Bronchoalveolar Lavage Fluid Findings, and Lung Morphology"

_nanomaterials, 2023, doi:10.3390/nano13182598_

Round 1
Reviewer 1 Report
This review represents a synthesis of the over all work of the authors team regarding the pulmonary toxicity and carcinogenicity of one MWCNT Mitsui-7. They compile all their acute, subchronic and chronic inhalation studies in rats in order to assess correlation between carbon nanotube lung burden and observed toxicity. As written by the authors there is a good correlation between these parameters for Mitsui-7. However, they do not really compare the data obtained with this MWCNT with data from the litterature obtained with other carbon nanotubes or cite/discuss studies which have performed such analysis. Mitsui-7 is the only CNT classified 2B by IARC, the others were not classified because of the lack of study at the time of the expertise. It would be nice to give some thought about the fact that long CNTs could be (or not) considered as potentially carcinogenic in rodents.
The authors discuss about how Mitsui-7 could impair macrophages mobility/activity/function, since this could be considered as a key event in tumor induction, the author should try to set up some kind of AOP to explain how Mitsui-7 could induce lung cancer.
The authors cite figures from previously published papers from their team and cite figures (line 725 Fig 9 from ref 50 and Line 264 Fig 5 from ref 15). It appears somewhat confusing, and one may argue that a reader would prefer have all the information needed in the same paper rather jumping from one paper to another to check the figures cited. These sentences should be written differently to mention papers bu not specific figures from these papers.
In some figures, data seem to have been log transformed but its seems this information does not appear in the legend and/or the text.
The authors should check figure 9, it seems to be inaccurate (X axis for male and lack of tumor for females at 0.2 mg/m3).
Minor comments: - Line 82 "TG-39" should be replaced by "GD-39"; Line 329 "reveled" should be replaced by "revealed". Ethical statement regarding animal experiments seem to be missing.
Author Response
This review represents a synthesis of the over all work of the authors team regarding the
pulmonary toxicity and carcinogenicity of one MWCNT Mitsui-7. They compile all their acute,
subchronic and chronic inhalation studies in rats in order to assess correlation between carbon
nanotube lung burden and observed toxicity. As written by the authors there is a good correlation
between these parameters for Mitsui-7.
However, they do not really compare the data obtained with this MWCNT with data from the literature obtained with other carbon nanotubes or cite/discuss studies which have performed such analysis.
No other 2-year inhalation studies have been performed.
There have been 2-year studies that administered MWCNTs by instillation into the lung. While we believe that the results of these instillation studies are reasonable, it is beyond the scope of this review to assess these studies beyond what we present in the Discussion section of our manuscript. It is also beyond the scope of this manuscript to assess the reliability of short term studies as predictors of long term inhalation exposure.
Mitsui-7 is the only CNT classified 2B by IARC, the others were not classified because of the lack of study at the time of the expertise. It would be nice to give some thought about the fact that long CNTs could be (or not) considered as potentially carcinogenic in rodents.
While no other 2-year inhalation studies have been performed, all CNTs that have been tested in 2 year instillation studies have been carcinogenic in the rat lung.
The proposed mechanism of MWNT-7 carcinogenicity is likely to function for other CNTs, and in our short summary of the pathway by which MWNT-7 induces lung cancer we state "This pathway suggests that any inhaled carbon nanotube that induces impaired macrophage motility is a potential carcinogen."
The authors discuss about how Mitsui-7 could impair macrophages mobility/activity/function,
since this could be considered as a key event in tumor induction, the author should try to set up
some kind of AOP to explain how Mitsui-7 could induce lung cancer.
We have added a paragraph to the end of the Discussion Section 4.3 summarizing a proposed pathway by which MWNT-7 induced lung carcinogenesis in our inhalation studies.
The authors cite figures from previously published papers from their team and cite figures (line
725 Fig 9 from ref 50 and Line 264 Fig 5 from ref 15). It appears somewhat confusing, and one
may argue that a reader would prefer have all the information needed in the same paper rather
jumping from one paper to another to check the figures cited. These sentences should be written
differently to mention papers but not specific figures from these papers.
We rewrote the Results section so that all of the data is presented in the present manuscript. However, we believe that referring to a specific figure in a cited manuscript in the Discussion section is reasonable and enables the reader to easily find the information we are referring to. Of course, if the Editor determines that we should remove "Figure 9" from the sentence "These studies are summarized in Figure 9 by Hojo et al., 2022 [50]" we will do so.
In some figures, data seem to have been log transformed but its seems this information does not
appear in the legend and/or the text.
It is generally not needed to indicate the scale used in a figure is a Log.
We would like to leave the decision of adding this information to the figure to the Editor.
The authors should check figure 9, it seems to be inaccurate (X axis for male and lack of tumor
for females at 0.2 mg/m3).
We thank the reviewer for this comment. The figure has been corrected.
Minor comments: - Line 82 "TG-39" should be replaced by "GD-39"; Line 329 "reveled" should be
replaced by "revealed".
We have amended the manuscript according to the reviewer's comment.
Ethical statement regarding animal experiments seem to be missing.
The ethical statement regarding animal experiments is in the first section of Methods.
Reviewer 2 Report
This manuscript reviewed the carbon nanotubes exposure to lung function, and in this manuscript, the authors also included some additional unpublished data in different time inhalation exposure studies. However, there are some concerns need to be addressed.
1. Many language errors should be corrected, there must be a space before the parentheses.
2. In the unpublished data, the correlation analysis should be supplied the statistical method, Person or others.
3. In Figure 3-1, the data should be compared with normalized standard.
4. Statistical analysis with a p-value should be included in correlation analysis.
5. In Figure 4 and Figure 6, the scale bar should be clearly showed in images.
Academic expression should be further polished in the revised manuscript.
Author Response
- Many language errors should be corrected, there must be a space before the parentheses.
We have carefully gone over the manuscript and corrected all errors in English spelling, grammar, and diction.
- In the unpublished data, the correlation analysis should be supplied the statistical method,
Person or others.
- In Figure 3-1, the data should be compared with normalized standard.
We apologize to the reviewer. We do not understand this comment. There is no normalized standard for lung burden versus total protein in the BALF.
- Statistical analysis with a p-value should be included in correlation analysis.
We apologize to the reviewer. We do not understand this comment. The p-value is the probability that the null hypothesis (r = 0) is true. However, the data presented in the scatter plot rejects the null hypothesis and consequently, we believe that addition of a p value could be misleading.
We would like to leave the decision of adding this information to the Figures/Table 2 to the Editor.
- In Figure 4 and Figure 6, the scale bar should be clearly showed in images.
Figures 4 and 6 have been amended according to the reviewer's comment.
Academic expression should be further polished in the revised manuscript.
We have carefully gone over the manuscript and corrected all errors in English spelling, grammar, and diction. We have also make extensive revision to the Results and Discussion sections.
Round 2
Reviewer 1 Report
The reviewer would like to thank the authors for their answers to the questions/comments and the corrections to the manuscript.
One last comment: Figure 10, the authors talk about tumor incidence. According to this figure, there is no tumor incidence in female exposed to 0.2 mg/m3 of MWCNT. Do the authors only take into consideration carcinomas? Perhaps the authors should mention the type of tumor in the legend of the figure.
Author Response
Yes, the reviewer is correct, Fig. 10 shows the incidence of carcinomas.
We have changed the figure and the legend to reflect the fact that only Lung Carcinoma incidence is shown in Fig. 10.
Reviewer 2 Report
The authors have not addressed my concerns about the statistical informations, this manuscript is not a pure review, therefore, I think the statistical information should be supplied in detailed.
Some errors had been corrected.
Author Response
We added a Statistics section to the Methods (p 6).
Reviewer 2 also commented that Minor editing of English language required, and "Some errors had been corrected."
We are not sure what this means. The manuscript has been reviewed by a native English speaker, and we believe that errors in English spelling, grammar, and diction have been corrected.